

# A Relaxed Eddy Accumulation (REA) LOPAP-System for Flux Measurements of Nitrous Acid (HONO)

Lisa von der Heyden[1], Walter Wißdorf[1], Ralf Kurtenbach[1], Jörg Kleffmann[1]

[1] Department of Physical and Theoretical Chemistry, Faculty for Mathematics and Natural Sciences, University of Wuppertal,
42097 Wuppertal, Germany

*Correspondence to*: Jörg Kleffmann (kleffman@uni-wuppertal.de)

**Abstract.** In the present study a Relaxed Eddy Accumulation (REA) system for the quantification of vertical fluxes of nitrous acid (HONO) was developed and tested. The system is based on a three-channel-LOPAP instrument, for which two channels are used for the updrafts and downdrafts, respectively, and a third one for the correction of chemical interferences. The
instrument is coupled to a REA gas inlet, for which an ultrasonic anemometer controls two fast magnetic valves to probe the two channels of the LOPAP instrument depending on the vertical wind direction. A software (PyREA) was developed, which controls the valves and measurement cycles, which regularly alternates between REA-, zero- and parallel ambient measurements. In addition, the assignment of the updrafts and downdrafts to the physical LOPAP channels is periodically alternated, to correct for differences in the interferences of the different air masses. During the study, only small differences
of the interferences were identified for the updrafts and downdrafts excluding significant errors when using only one interference channel. In laboratory experiments, high precision of the two channels and the independence of the dilution corrected HONO concentrations on the length of the valve switching periods were demonstrated.

A field campaign was performed in order to test the new REA-LOPAP system at the TROPOS monitoring station in Melpitz, Germany. HONO fluxes in the range of $-4 \cdot 10^{13}$ molecules m$^{-2}$ s$^{-1}$ (deposition) to $+1.0 \cdot 10^{14}$ molecules m$^{-2}$ s$^{-1}$ (emission) were
obtained. A typical diurnal variation of the HONO fluxes was observed with low, partly negative fluxes during night-time and higher positive fluxes around noon. After an intensive rain period the positive HONO emissions during daytime were continuously increasing, which was explained by the drying of the upper most ground surfaces. Similar to other campaigns, the highest correlation of the HONO flux was observed with the product of the NO$_2$ photolysis frequency and the NO$_2$ concentration ($J(NO_2) \cdot [NO_2]$), which implies a HONO formation by photosensitized conversion of NO$_2$ on organic surfaces,
like e.g. humic acids. Other postulated HONO formation mechanisms are also discussed, but are ranked being of minor importance for the present field campaign.



## 1 Introduction

During the last 25 years, high nitrous acid (HONO) mixing ratios have been observed during daytime under very different
environmental conditions pointing to a major contribution of HONO photolysis to the oxidation capacity of the lower
atmosphere (Neftel et al., 1996; Zhou et al., 2002a; Kleffmann et al., 2005; Acker et al., 2006; Elshorbany et al., 2009; Villena
et al., 2011; Li et al., 2012; Yang et al., 2014; Hou et al., 2016; Lee et al., 2016; Tan et al., 2018; Slater et al., 2020). These
high HONO levels can only be explained by strong daytime sources, for which (i) heterogeneous reduction of nitrogen dioxide
($NO_2$) in the presence of organic photosensitizers (George et al., 2005; Stemmler et al., 2006; 2007; Sosedova et al., 2011; Han
et al. 2016a; 2016b; 2017; Yang et al., 2021a), (ii) heterogeneous photolysis of nitric acid/nitrate (Zhou et al., 2003; 2011;
Laufs and Kleffmann, 2016; Ye et al., 2016; 2017; Romer et al., 2018; Shi et al., 2021) and (iii) bacterial production of nitrite
in soil (Su et al., 2011; Ostwald et al., 2013; Maljanen et al., 2013; Oswald et al., 2015; Scharko et al., 2015; Weber et al.,
2015) and/or desorption of adsorbed HONO from soil surfaces during daytime (Donaldson et al., 2014; VandenBoer et al.,
2014; 2015) have been identified. In contrast, other proposed sources, like the gas-phase reaction of excited $NO_2$ with water
(Crowley and Carl, 1997; Li et al., 2008; Carr et al., 2009; Amedro et al., 2011), the photolysis of nitro-phenols or similar
compounds (Bejan et al., 2006; Yang et al., 2021b) and the gas-phase reaction of $HO_2 \times H_2O$ complexes with $NO_2$ (Li et al.,
2014; 2015; Ye et al., 2015) are of minor importance. Except proposed HONO formation by particle nitrate photolysis (Ye et
al., 2016; 2017), mainly ground surface sources have yet been identified in laboratory and field studies to explain atmospheric
HONO formation during daytime in the atmosphere (Kleffmann, 2007). This is in good agreement with recent HONO gradient
studies during daytime by the MAX-DOAS technique (Garcia-Nieto et al., 2018; Ryan et al., 2018; Xing et al., 2021).

In most field studies, the daytime HONO source strength was determined from HONO levels exceeding theoretical photo-
stationary state (PSS) values, for which correlations of the daytime source with the photolysis rate coefficient $J(NO_2)$ or the
irradiance and the $NO_2$ concentration were often observed (Vogel et al., 2003; Su et al., 2008; Elshorbany et al., 2009; Sörgel
et al. 2011; Villena et al., 2011; Wong et al., 2012; Lee et al., 2016; Crilley et al., 2016). In these studies, the HONO source is
mathematically treated as a gas phase process, despite its heterogeneous origin. Thus, the quantification of the daytime source
is erroneous and depends on the height of the measurements above the ground and the vertical mixing of the atmosphere. In
addition, the assumed PSS conditions may also not be fulfilled when HONO and its precursors are measured close to their
sources (Lee et al., 2013; Crilley et al., 2016).

In contrast, flux measurements are able to give direct information about ground surface production and deposition and are a
better tool to quantify ground sources of HONO in the lower atmosphere. Available flux observations indicate different HONO
precursors. Harrison and Kitto (1994), Ren et al. (2011) and Laufs et al. (2017) found a relationship of the HONO flux with
the $NO_2$ concentration and also its product with light intensity, which can be explained by the photosensitized conversion of
$NO_2$ on humic acid surfaces (Stemmler et al., 2006). In contrast, for grassland spread with manure upward HONO fluxes could
not be explain by an $NO_2$ driven mechanism (Twigg et al., 2011). For high nitrogen soil content, e.g. after fertilization, up to
two orders of magnitude higher HONO fluxes compared to most other studies were recently observed in soil chambers (Xue
et al., 2019; Tang et al., 2019). Thus, the latter three studies point to a soil nitrogen driven HONO formation mechanism. In
contrast, Zhou et al. (2011) observed a correlation of the HONO flux with adsorbed nitric acid and short wavelength radiation,
which was explained by photolysis of nitric acid adsorbed on canopy surfaces. Finally, for flux measurements in a forest
clearing dominant formation processes explaining the positive daytime HONO fluxes remained unclear. In the same study at
a co-located forest floor, only net HONO deposition was observed at the much lower irradiance levels compared to the forest
clearing (Sörgel et al., 2015). The difference could be explained again by a photolytic HONO formation mechanism. In
conclusion, based on available HONO flux studies, the origin of the main ground surface daytime HONO source is still
controversially discussed.

Nowadays, eddy covariance (EC) is the most commonly applied method to measure fluxes between the surface and the
atmosphere. The lack of HONO instruments, however, which are fast and sensitive enough for the EC method, requires the





use of indirect methods like the aerodynamic gradient (AG) method (Harrison and Kitto, 1994; Twigg et al., 2011; Sörgel et al., 2015; Laufs et al., 2017) or the Relaxed Eddy Accumulation method (REA) (Ren et al., 2011; Zhou et al., 2011; Zhang et al., 2012).

In the present study a Relaxed Eddy Accumulation system for the quantification of vertical fluxes of HONO based on the
LOPAP technique was developed and tested in the laboratory and in the atmosphere.

## 2    The REA-Method

Turbulent transport is the most important process for the exchange of energy and trace gases between the ground and the atmosphere. When only slow instruments are available, the REA-method is used to determine fluxes of trace gases, for which air masses are collected and analysed in two channels depending on the sign of the vertical wind ($w$) (Businger and Oncley,
1990). According to meteorological convention, positive and negative vertical wind directions reflect updrafts and downdrafts, respectively. Controlled by the vertical wind data measured by a 3D anemometer, fast valves are used to separate the different air masses to feed the two channels of the instrument. With a constant sample flow rate, the flux $F_c$ of a trace gas $c$ is calculated according to equation (1) from the mean concentration difference of the two channels $\left(\overline{c_{up}} - \overline{c_{down}}\right)$, the standard deviation of the vertical wind speed $\sigma_W$ and a coefficient $b_0$, which is 0.627 under ideal joint Gaussian distribution of $w$ and $c$ (Wyngaard
and Moeng, 1992; Ammann and Meixner, 2002; Sakabe et al., 2014):

$$(1)\qquad F_c = b_0 \cdot \sigma_w^{30\,Min} \cdot \left(\overline{c_{up}} - \overline{c_{down}}\right).$$

An averaging interval of typically 30 min is chosen to ensure that all eddies of different sizes and frequencies are captured, which contribute to the turbulent transport. Several field studies have shown that, depending on the stability of the atmosphere, the coefficient $b_0$ deviates from the ideal value and can be in the range of 0.51-0.62 (Baker et al., 1992; Katul et al., 1996;
Ammann and Meixner, 2002).

To increase the measured concentration difference and to improve the signal-to-noise ratio, the use of a deadband is recommended (Businger and Oncley, 1990; Oncley et al., 1993). Here, sample air is collected only when the vertical wind speed is outside a defined deadband of the width $\pm w_0$ around zero, while smaller eddies with low energy are not accounted for. Besides decreasing the flux errors, the use of a deadband also conserves the used valves by reducing the valve switching
frequency. The half width of the deadband $w_0$ is determined from the variance of the vertical wind $\sigma_w^{5\,Min}$ during the past 5 min:

$$(2)\qquad w_0 = K \cdot \sigma_w^{5\,Min}.$$

Depending on the turbulence distribution and the concentration of $c$, $K$-values in the range 0.5-1 are used. Equation (1) can be reformulated when a deadband is used:

$$(3)\qquad F_c = b \cdot \sigma_w^{30\,Min} \cdot \left(\overline{c_{up\,(w > w_0)}} - \overline{c_{down\,(w < -w_0)}}\right).$$

Often variable $b$-parameters are calculated for each averaging interval, for which different methods have been developed. By the proxy method, a passive scalar $s$ (e.g. temperature) is measured simultaneously and the flux of this scalar is determined by means of the Eddy Covariance method (see e.g. Aubinet et al., 2000). $b_{Proxy}$ is then calculated as:

$$(4)\qquad b_{Proxy} = \frac{F_S}{\sigma_w^{30\,Min} \cdot \left(\overline{s_{up(w > w_0)}} - \overline{s_{down(w < -w_0)}}\right)}.$$

By the proxy method, scalar similarity of the transported scalar $s$ and trace gas $c$ is assumed, which implies a similar transport efficiency for all sizes of eddies. This assumption is, however, not always fulfilled (Ruppert et al., 2006). Alternatively, $b_w$ can be also calculated by the vertical wind statistics (Baker et al., 1992; Baker, 2000):



(5) $\quad b_w = \dfrac{\sigma_w^{30\,Min}}{\left(\overline{w}_{up(w\,>\,w_0)} - \overline{w}_{down(w\,<\,-w_0)}\right)}$ .

The value of $b$ decreases when using a deadband. Simulation studies of turbulence data have consistently shown that $b_{model}$ is

exponentially approaching a limit value when increasing the width of the deadband (Pattey et al., 1993; Katul et al., 1996):

(6) $\quad b_{model} = \left(1 - a \cdot \left[1 - e^{\left(-d \cdot \frac{w_0}{\sigma_w}\right)}\right]\right) \cdot b_0$ .

The parameter $b_0$ describes $b$ without using a deadband and $a$ and $d$ are empirical coefficients ($a = 0.37$ and $d = 1.958$) in the

nonlinear regression model describing $b$ as a function of the normalized half width of the deadband ($w_0/\sigma_W$) (Pattey et al.,

1993).

To exclude systematic errors in the assignment of the air masses to the two channels, an essential requirement for the

application of equation (3) is a mean vertical windspeed $\overline{w}$ of zero in each averaging interval. This requirement can be violated,

e.g. when the investigated surfaces are not perfectly horizontal, or when the anemometer is not perfectly aligned. This is

corrected for by rotation of the three-dimensional wind data $u$, $v$, and $w$ in a right-handed Cartesian coordinate system, which

follows the physical streamlines (double rotation method, see Kaimal and Finnigan, 1994). Using a back-looking moving

average window of 5 minutes, the rotation is done immediately during data acquisition.

### 3    Set-up of the instrument

#### 3.1    Modified LOPAP instrument

For the REA system developed in the present study the LOPAP technique is used, which is explained in detail elsewhere

(Heland et al., 2001; Kleffmann et al., 2002). HONO is sampled in a temperature-controlled stripping coil by a selective

reaction with the sampling reagent (R1: 1 g $l^{-1}$ sulfanilamide in 1 N HCl) forming a diazonium salt. In the present study, the

originally recommended sulfanilamide concentration of 10 g $l^{-1}$ (Heland et al., 2001) was significantly reduced leading to an

improved signal stability and a still almost quantitative sampling efficiency of 99.6 % at a gas flow rate of 1.05 $l$ min$^{-1}$ (298.15

K, 101325 Pa) applied in the present study. The more stable signal is explained by prevention of sulfanilamide crystal formation

at the inlet of the stripping coil, especially at low humidity.

The stripping coil is installed in an external sampling unit directly in the atmosphere of interest, by which sampling artefacts

in sampling lines are minimized. The sampling solution is transferred by a 3 m long temperature-controlled reagent line to the

LOPAP instrument, where it reacts with the second reagent (R2: 0.1 g $l^{-1}$ N-(1-naphthyl)-ethylenediamine-dihydrochloride

(NEDA) in water) forming a strongly absorbing azo-dye. The absorbance of the dye is determined by modified Lambert-Beer's

law (Heland et al. 2001) in a *liquid-core-waveguide*, which consists of a specific Teflon tube (Teflon AF 2400, 0,6 mm i.d.,

195 cm length) with a lower refractive index (n = 1.29) than the reagents. For the correction of interferences two channels in

series are used. In the first coil, HONO is almost completely sampled in addition to small quantities of interfering species. In

contrast, only the interferences are sampled in the second coil. From the difference of both channels, corrected for the sampling

efficiency, the HONO concentration is determined (Kleffmann and Wiesen, 2008). The interference correction assumes a small

sampling efficiency of the interfering species, which was confirmed for many trace gases (Heland et al., 2001; Kleffmann et

al., 2002). In addition, intercomparison with the DOAS technique showed excellent agreement under complex photosmog

conditions in a smog chamber and in the urban atmosphere (Kleffmann et al., 2006).

For the application in a REA system, ideally four channels would be necessary to quantify HONO and interferences in the

updrafts and downdrafts. However, this was not possible in one housing of a standard LOPAP instrument. On the other hand,

two separate LOPAP instruments would not fit into the available field rack (see section 3.5) and the setup would become too

bulky for turbulence measurements. Thus, a three channel LOPAP-instrument was developed, in which a double-stripping coil

(channel 1 and 2) is used to probe HONO and interferences in one air mass as explained above. An additional single stripping





coil (channel 3) probes the other air mass (see Figure 1 and Figure S1). The measured signal in channel 2 is used for interference correction of both channels 1 and 3, which assumes that the concentrations of the interfering species is almost similar for the updrafts and downdrafts. This was confirmed in all measurements in the ambient atmosphere (see sections 4.1 and 5.1).

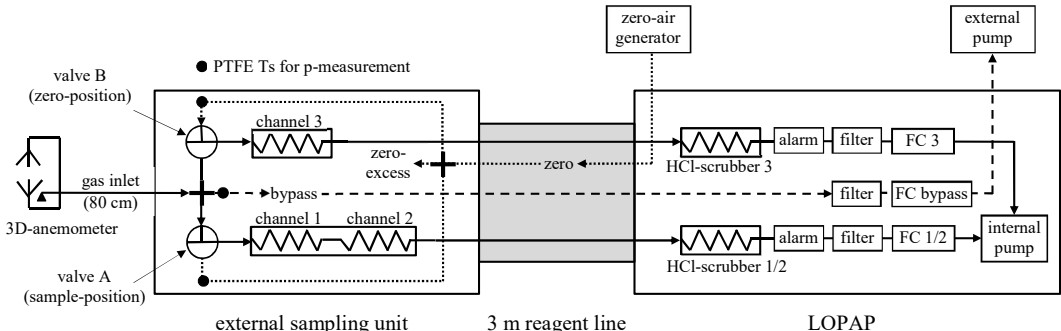


**Figure 1: Gas flow scheme of the three-channel REA-LOPAP.**

For the additional implementation of the REA gas inlet (see next section) a new compact sampling unit was developed (see Figure S1). In addition, the temperature-controlled reagent line was modified, housing 9 liquid lines for the three channels (see Figure S2) and additional gas lines for the bypass and the zero-gas generator (see Figure 1). In the LOPAP instrument an

additional 2 $l$ min$^{-1}$ mass flow controller (Bronkhorst), a second HCl-scrubber and security bottle ("alarm") were installed for the new channel 3. In addition, a 5 $l$ min$^{-1}$ mass flow controller (Bronkhorst) and an external membrane pump (KNF) were installed to control the bypass flow of the REA gas inlet (see Figure 1). Besides, also the liquid flow scheme was adopted to the new 3-channel LOPAP (see Figure S2). Since the internal peristaltic pump (Ismatec) has only 16 channels, from which already 13 are used in a normal 2-channel LOPAP, an additional external 4-channel peristaltic pump was necessary to remove

the bubble-containing waste of the three channels from the glass T-junctions behind the stripping coils (see Figure S2).

## 3.2   The REA gas inlet

For the REA-method two separate channels are probed with updrafts and downdrafts by switching two valves, which are placed in the external sampling unit (see Figure 1 and Figure S1). Caused by its size and to minimize its impact on the turbulence measurements of the ultrasonic anemometer (CSAT3B, Campbell Scientific Inc.) the external sampling unit is

installed ~80 cm leeward to the centre of the anemometer (see Figure S4). The ~80 cm long PFA inlet line (4 mm i.d.) from the anemometer to the external sampling unit was covered by aluminium foil to minimize artificial photochemical formation of HONO (Zhou et al., 2002b). The inlet was protected against rain by a small cone (see Figure S4, right) and was positioned below the anemometer, slightly shifted to the lee side, following the recommendations of Kristensen et al. (1997). The use of an inlet line is normally not recommended for the LOPAP technique, but was necessary to spatially separate the anemometer

and the external sampling unit. Thus, small positive artefacts could be possible by heterogeneous dark formation of HONO in the inlet. However, this should not affect the HONO fluxes, which are calculated from the concentration difference of updrafts and downdrafts and not from the absolute HONO levels (see equation (3)).

During REA measurements each channel of the instrument is fed for a certain time with ambient air, depending on the vertical wind direction. In contrast, for closed valves, when the vertical wind is too small during deadband conditions or during zero

measurements, the LOPAP channels are operated by zero-air. Therefore, fast (<12 ms response time) three-way valves (miniature inert PTFE isolation valves series 1, Parker Hannifin Corp., USA) were used to feed the stripping coils either with ambient air or with zero-air. By using a PTFE cross (BOLA, 1.6 mm i.d.), the sample air from the PFA inlet line is directed to the two valves (see Figure 1 and Figure S1). Since the inlet line has to be flushed by ambient air during the deadband to



minimize heterogeneous HONO formation, the remaining open port of the PTFE cross is connected to the bypass line (see

Figure 1 and Figure S1) for which a flow rate of 2.65 $l$ min$^{-1}$ (298.15 K, 101325 Pa) is used in the present study.

The zero-air lines of both channels are connected via a stainless-steel cross (Swagelok, Ohio) to the main zero-gas line. The remaining open port of the cross is used for the excess vent of the zero-air. Zero air was obtained from a home-made zero-air generator (see Figure S4), where compressed ambient air by a membrane pump is pushed through a particle filter, a flowcontroller (Bronkhorst: 5 $l$ min$^{-1}$), two cartridges filled with active charcoal and active charcoal coated with $Na_2CO_3$ and

another particle filter. The zero-air flow is adjusted slightly higher than the gas flow of both channels, when both valves are closed during deadband and zero-air periods. The dilution of the ambient air during REA measurements is later corrected for during data evaluation (see section 4.1).

During REA-measurements both valves switch between ambient and zero-air. For both measurement conditions a constant air flow through the stripping coil is assumed to correctly calculate the dilution of the ambient air in each channel during the

averaging interval. However, constant air flow can only be obtained when the pressure at the inlet of the stripping coil is independent of the valve position. Initially, this was not the case since the pressure at the inlet of the stripping coil was lower during ambient air compared to zero-air measurements. This was caused by the longer inlet line and the higher gas flow rate including sample and bypass flow compared to the flow rate in the shorter zero-air line from the stainless-steel cross (=ambient pressure) to the valve (see Figure 1). Therefore, during first test measurements with variation of the valve switching times (see

section 4.2) systematic and significant variations of the resulting dilution-corrected HONO concentrations were observed. Since the flow controllers of the two channels (valve A/channels 1+2; valve B/channel 3) are installed inside the LOPAP instrument and are connected via particle filter, security bottle, HCl-scrubber and a more than 3 m long gas line to the stripping coil (see Figure 1), the flow rates were changing after switching the valves until new pressure equilibria were reached in the volumes from the stripping coil to the flow controller. This problem appeared particularly with short valve switching periods

as applied during REA measurements.

To avoid this inlet pressure problem, first, the diameter of the PFA inlet line was increased from initially 1.6 mm i.d. to 4 mm i.d., decreasing the depression at the inlet of the stripping coils. Second, the length of the 1.6 mm i.d. PFA zero lines from the stainless-steel cross (= ambient pressure) to the valves was increased until the pressure at the valve was independent of the valve position (depression similar during ambient and zero-measurements). To adjust the pressure in the REA inlet, three

PTFE T-junctions (Bola, 1.6 mm i.d.) were installed at the end of the two zero-air lines at the valves and at the inlet of the bypass line (see Figure 1 and Figure S1). After removing the blind caps from the Ts, the pressure can be measured by a pressure gauge (baratron 0-1000 hPa). For fine adjustment of the pressure, the bypass flow rate is adjusted, which has only an influence on the depression during ambient air measurement, but not during zero-air supply (see Figure 1). For the fine adjustment of the pressure during deadband – here both channels are operated under zero-air and the pressure must be similar at the inlet of

both coils – the sample flow rates of both channels were slightly varied. The exact adjustment of the inlet pressures at the two stripping coils under the different operation conditions was the largest problem during the development of the REA-LOPAP system. Only after this was solved, successful REA-measurements were possible.

### 3.3     REA data logging and valve controlling software - PyREA

Data-logging, processing of the anemometer raw data and control of the REA valves and measurement cycles was performed

with a control program, *PyREA*, developed primarily in the *Python* programming language. The applied REA theory and the implemented formula are explained in section 2. Specifically, equation (5) was used to calculate $b_w$, which is later used to calculate the HONO flux by equation (3). Details of the program are explained in the supplement section S3.

The software was operated on a single board computer (Raspberry Pi, version 3 B+). The user interface of *PyREA* was accessed via a Secure Socket Shell (SSH) connection from the LOPAP data acquisition computer, which allows to control the REA

parameters (e.g. mode switching times). Here, the sequence of REA-, zero- and parallel ambient measurements can be adjusted



to run automatically ("auto-REA mode"). During REA measurements, each valve switches between ambient air and zero-air, controlled by the vertical wind signals from the anemometer and the width of the deadband. Regular zero-measurements (both valves are closed) are necessary for the LOPAP technique (Heland et al., 2001) and regular parallel ambient measurements (both valves are open) were used to improve the precision between the two channels for the updrafts and downdrafts.

In the software the different lag times when switching from ambient air measurements to dead-band (sample + bypass inlet flow) or from dead-band to ambient measurement (only bypass inlet flow) can be adjusted for the REA measurements. Besides the flow rates and the volume of the inlet (10.1 cm³), also the response time of the valves (12 ms) and the delay of the anemometer (95 ms: only first 10 ms sample of each 100 ms interval is provided by the anemometer) have to be considered. The lag time is calculated by the variable inlet residence time minus the anemometer delay time and the valve response time.

Alternatively, the valve can also be operated manually by the graphical interface (GUI), which is necessary during adjustment of the pressure in the REA inlet for example (see section 3.2). Furthermore, different test modes were implemented in the software, which were used during the development of the REA-system. For example, artificial anemometer data can be simulated for tests inside the laboratory (test-mode: *simulate_anemometer*), or the valves can be operated in a pre-defined continuous sequence (test-mode: *valve_function_test*). The latter was used to test the dilution correction for variable valve
switching times (see section 4.2).

### 3.4    Laboratory set-up

To test the new REA-system in the laboratory and in front of the facade of the laboratory building, a mobile laboratory rack was developed, on which all components of the system were installed. The external sampling unit and the anemometer were fixed on an aluminium arm with a similar geometry compared to the field set-up (see Figure S4 left). Through an opening in
an exchanged window element of the facade of the laboratory the external sampling unit and the anemometer could be moved outside of the building, while the rack with the LOPAP being inside the laboratory (see Figure S4 right). The opening could be closed by two PVC-plates with smaller holes for the aluminium arm, the insulated reagent line from the external sampling unit to the LOPAP instrument and the cables from the anemometer. This set-up allowed to start measurements inside the laboratory, which was necessary for gas flow rate measurement during calibration for example. Then, with the running
instrument the set-up could be changed to ambient test measurements in front of the facade of the building.

### 3.5    Field set-up in Melpitz

For the field campaign in Melpitz a field rack was available, which was already used in our former gradient measurements in Grignon, France (Laufs et al., 2017). For the present study, the field rack was upgraded with an external air conditioning system with 1 kW cooling power (SoliTherm Outdoor, Seifert Systems GmbH, see Figure S5, left) to ensure constant
temperatures inside the rack also under sunny summertime conditions.

The field campaign took place at the field site Melpitz (12.9277° E., 51.5255° N., 86 m NN), which is operated by the TROPOS institute in Leipzig, Germany and is used as a regional station by the Global Atmosphere Watch (GAW) program. The station is located on grass land, which is surrounded by farmland. The nearest buildings of the small village Melpitz are located ~360 m to the east of the station. The railroad tracks from the line Leipzig – Torgau and the state road 87 are located ~1 km
and ~1.5 km north of the station, respectively. The next buildings in south-westerly direction are from the village Klitzschen at ~1.5 km distance. A small district road connecting Melpitz and Klitzschen is located ~400 m south of the station. The REA-system was installed ~80 m west of the TROPOS measurement containers and for typical south-westerly winds, no influence of the station on the turbulence and HONO measurements is expected. At ~130 m distance to the west there is a single open line of smaller trees and bushes (see Figure S5, right) with is converging to ~70 m to the south of the field site. The REA-
sampling unit and the anemometer were installed on a small mast in south-westerly direction (220°) to the field rack facing to the expected main wind direction and thus minimizing the influence of the field rack to the turbulence measurements (see





Figure S5). The centre of the anemometer was located at ~200 cm height above the grass surface. To avoid turbulence disruptions by the inlet line, the gas inlet was slightly shifted to the lee side (8 cm) and placed below the anemometer at ~170 cm height, following the recommendations of Kristensen et al. (1997). During the campaign, regular parallel ambient
measurements of 30 min were used after two measurement cycles (one measurement cycle consists of a 240 min REA period and a subsequent 30 min zero period). As filter for the deadband (see section 2) a value of $K = 0.9$ was used in the first half of the campaign. In contrast, a value of $K = 0.6$ was used during the second week, which will be mainly presented here.

Besides the HONO flux measurements, also the actinic flux and the photolysis frequencies $J(NO_2)$, $J(HONO)$ and $J(O^1D)$ were determined by a spectroradiometer with a cooled CCD detector and a $2\pi$-collector (Meteoconsult) on the roof of one
measurement container at the main TROPOS station. In addition, NO- and $NO_2$-concentrations were measured at the TROPOS station by the chemiluminescence technique (Eco Physics, CLD 770 AL) with blue light $NO_2$-converter (homemade, University of Wuppertal). The instrument was calibrated six times during the campaign with an NO calibration gas (Messer 450±10 ppb) for which also the $NO_2$ converter efficiency was determined to be (50.4±0.7) % by an $O_3$-titration unit (Anysco GPT). For the $NO_x$ measurements the common gas inlet of the standard $NO_x$ instrument from the TROPOS station was used,
the data of which was not considered here, caused by its higher precision errors and detection limit.

Additionally, data for nitric acid ($HNO_3$) measured by a MARGA instrument (Stieger et al., 2018), wind-direction and -speed, temperature and relative humidity from the TROPOS station were used for data evaluation.

## 4    Laboratory tests

With the laboratory set-up (see section 3.4) test measurements were performed inside the laboratory and at the facade of the
laboratory building to develop a data evaluation procedure (section 4.1) and to validate the dilution correction for variable valve switching times (section 4.2).

### 4.1    Data evaluation

During flux measurements simultaneous data from the data acquisition system on the LOPAP computer (HONO raw absorbance) and from the PyREA software on the Raspberry Pi (three-dimensional windspeed ($u$, $v$, $w$), air temperature ($T$)
and valve switching data from the REA inlet) are collected. The software also calculates wind direction and windspeed, as well as the REA parameters $b$ and $\sigma_w$.

For both data logging systems 30 s data are recorded, for which the REA data are calculated both, as 5 min and 30 min running averages (see supplement section S3), which are first combined in one data file. Since the time correction of the LOPAP data (lag time between sampling and dye detection) show a continuous shift caused by a slight reduction of the liquid flow rates of
the peristaltic pump, the time-corrected 30 s LOPAP data are synchronized to the 30 s REA data.

The LOPAP data are evaluated similar to the usual procedure (see Heland et al., 2001; Kleffmann et al., 2002). Here, time response (time for 10-90 % increase of the signal: ~5 min) and time correction (time to 50 % change: ~16 min) are determined from the raw data ($\log(I_{ref}/I)$, see Heland et al., 2001) and the time stamps of the PyREA software. Furthermore, the raw absorbance of only the zero measurements is described by polynomials as a function of time, which are subtracted from the
raw absorbance data, leading to zero-corrected data in the form required by Lambert-Beer's law ($\log(I_0/I)$), which should be zero in between their precision errors during all zero measurements. Next, from the calibration of the instrument by a nitrite standard in R1 (0.01 mg $l^{-1}$) under zero air, the slopes of the analysis functions (ppt ABS$^{-1}$) are determined for all three channels, from which the diluted concentrations are calculated by multiplying with the zero-corrected absorbance data (see Figure 2).



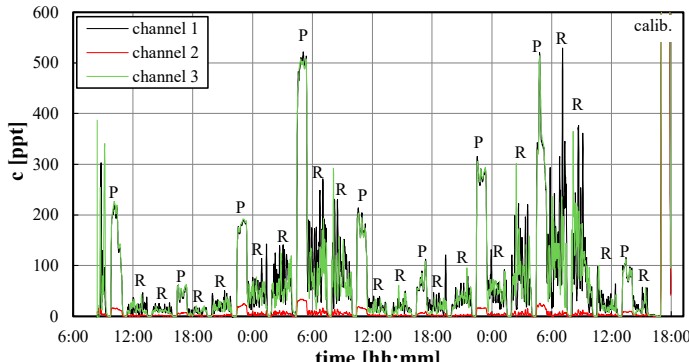

**Figure 2: Mixing ratios (30 s raw data) of the three LOPAP channels during test measurements at the university of Wuppertal. Periods named by „P" show undiluted parallel ambient measurements (60 min), while the REA-measurements (dilute samples) are signed by "R" (120 min). In between, regular zero-measurements (30 min) are performed.**

Since HONO fluxes are determined from small concentration differences between updrafts and downdrafts (see equation (3)), high precision is required for channels 1 and 3. For validation of this requirement, regular parallel ambient measurements are performed by the system, for which both channels measure exactly the same air masses. For the test campaign shown in Figure 2 excellent agreement between both channels was observed for all parallel ambient measurements (periods marked by "P"). In addition, a high correlation of both channels ($R^2 = 0.9986$), excellent agreement (slope: $0.9982\pm0.0014$) and an insignificant intercept of 2 ppt within the quantification error limit were observed. Since the ratio channel 1 / channel 3 showed a small but significant variation with time, the ratio was parameterized as a function of time for later harmonization of both channels (see below).

Next, the concentrations of all three channels were corrected for dilution during the REA-measurements by using the valve switching statistics recorded by the PyREA program for each averaging interval. The moving average data of 5 minutes were used for this correction due to the similar physical time response of the LOPAP instrument (see section 3.1). Then, the dilution-corrected concentrations were harmonized by the parameterization derived from the parallel ambient measurements (see above). All channels were corrected by half of the ratio (channel 1 / channel 3) determined during the parallel ambient measurements, for which channels 1 and 2 were divided and channel 3 multiplied by this ratio, respectively.

From the corrected and harmonized concentrations in channel 1 and 3 (HONO + interference) the dilution-corrected and harmonized concentrations in channel 2 (interference) were subtracted and HONO concentrations were calculated for both air masses considering the known HONO sampling efficiency of the stripping coils.

From the resulting HONO concentrations of both channels an average HONO concentration can be calculated, weighted by the fractions of sampling times of both channels during the 5 min averaging periods (see Figure 3). This average HONO concentration determined by the REA system should be similar to concentrations determined with a normal LOPAP instrument. Deviations caused by missing data during the deadband are considered small, since small concentration differences between the updrafts and downdrafts are expected during the low turbulence deadband conditions. This is confirmed by the non-significant variations of the HONO-concentrations when the system switches from parallel ambient measurements (continuous sampling of ambient air in both channels) to REA-measurements (see Figure 3). Thus, in addition to the HONO fluxes (see below) also HONO concentrations can be determined, similar to a normal LOPAP-instrument. It should be mentioned again that the average HONO concentration determined by the REA-LOPAP system may be affected by systematic artificial HONO formation in the inlet line. However, caused by the high air flow rate of 3.7 $l$ min$^{-1}$ and the short PFA inlet line (80 cm) covered by aluminium foil, artificial heterogeneous HONO formation in the dark inlet during the short residence time of 160 ms is considered small. An intercomparison with a normal LOPAP instrument without any inlet line is required to further confirm this assumption in the future.





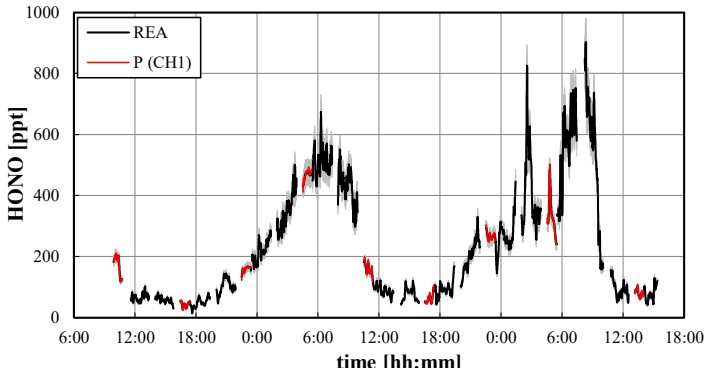

**Figure 3: Weighted average HONO concentration determined from both channels during REA measurements (REA) and HONO**
**concentration from channel 1 during parallel ambient measurements (P (CH1)) for the test campaign at the university of Wuppertal**
**(30 s data, 5 min running mean). The grey shaded area reflects the accuracy of the data.**

To validate that the applied interference correction using a three-channel system is applicable, the assignment of the updrafts and downdrafts to the two physical LOPAP channels is regularly alternated after each parallel ambient measurement by the PyREA software. In the mode "up", channels 1 and 2 (double stripping coil) are fed with ascending air and channel 3 (single

stripping coil) measures the descending air, while the assignment is inverted in the mode "down". Thus, the relative interference can be determined for both air masses. For the test measurements at the university of Wuppertal average interferences of (8.4±2.4) % and (7.5±4.3) % were determined for the updrafts and downdrafts, respectively. These values are similar within their combined variability. Even if the difference was considered, this would only affect the HONO concentrations in channel 3 by less than ±1 %, which is lower than the precision error. In the case of potential larger differences

of interferences in other campaigns, the data from channel 3 could be corrected considering the measured average ratio $interference_{up}/ interference_{down}$ depending on the mode ("up"/"down") applied.

Besides the validation of the interference correction method, the regularly changing assignment of the air masses to the two physical channels of the REA-LOPAP has to be considered for the determination of the HONO flux, which is calculated from the concentration difference of both channels ($\Delta HONO = HONO_{up} - HONO_{down}$). Due to the requirement of the REA-method

(see section 2) 30 min running means of $\Delta HONO$ are calculated. By using the measured ambient pressure and the air temperature $\Delta HONO$ is converted from mixing ratios [ppt] to concentrations [molecules m$^{-3}$]. According to equation (3) (see section 2) HONO fluxes [molecules m$^{-2}$ s$^{-1}$] are obtained by multiplication of $\Delta HONO$ with 30 min running mean values of $b$ [---] and $\sigma_w$ [m s$^{-1}$] obtained from the PyREA software. The fluxes determined during the test measurements in Wuppertal were however not further considered, caused by the distorted turbulence in front of the facade of laboratory building.

**4.2    Variation of valve switching periods**

After correctly adjusting the pressure in the REA inlet (see section 3.2) the valve switching periods were systematically varied by using the PyREA test-mode *valve_function_test* (see section 3.3). HONO measurements inside the laboratory were performed, for which HONO concentrations are changing slowly and a constant concentration can be assumed for a small span of time. A sequence of zero- (both valves closed), parallel ambient measurements (both valves open) and subsequent valve

switching periods was performed, for which constant valve switching times of 1, 2 and 3 s were applied for the periodically switching valve positions (1) up, (2) dead-band, (3) down and (4) dead-band, respectively. Caused by the chosen valve switching sequence, the HONO concentration is diluted exactly by a factor of four in both channels during these experiments (see Figure 4: "P" and "x/x/xs").





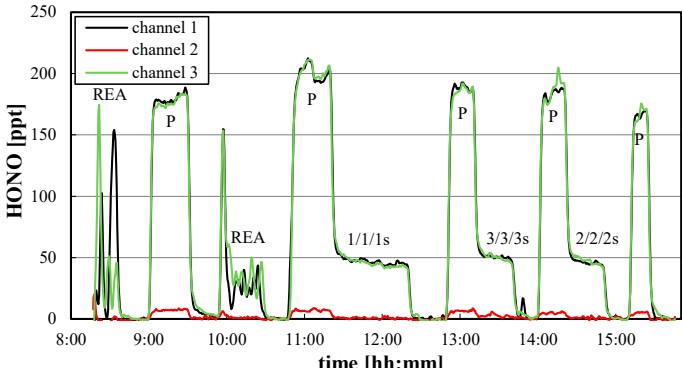

**Figure 4: HONO mixing ratios during test measurements using variable valve switching times (30 s data). „P": parallel ambient measurements; „x/x/xs": valve switching experiments with constant periods for up-, down and deadband measurements (each 1, 2, 3 s, sequence: (1) up, (2) dead-band, (3) down and (4) dead-band; dilution each by a factor of four).**

For the parallel ambient measurements an excellent agreement between channels 1 and 3 was again observed, with an average ratio (1/3) of 1.00±0.02. Thus, no further harmonization of the two channels (see section 4.1) was performed for the short measurement period applied. For the valve switching test the expected dilution by a factor of four could be confirmed in both channels. Here, a mean ratio (channel 1 / channel 3) for the dilution corrected data of 1.01±0.03 was observed. In addition, no significant differences between the HONO data during parallel ambient measurements and the dilution-corrected data during the valve-switching experiments were observed. For the average ratios (parallel ambient measurements / valve switching) values of 0.99±0.04 and 1.01±0.04 were observed for channels 1 and 3, respectively. Thus first, the agreement between both channels is independent of the valve switching period and second, the dilution is correctly determined by the system, which is the basic requirement for REA-measurements.

### 4.3    Flux errors

Errors in the fluxes were determined by error propagation of the accuracy of the LOPAP instrument (7 %), estimated uncertainty in the harmonization function derived from the parallel ambient measurements (5 %), the precision errors of the 30 min values of $\sigma_w / b_w$ (standard deviation of the 5 min running mean data) and the HONO concentration difference ΔHONO. For the latter the precision of the LOPAP instrument (2.2 % for the campaign in Melpitz), the detection limit (0.8 ppt for the campaign in Melpitz) and the dilution ratios during REA-measurements were considered for all channels of the instrument by error propagation. Possible uncertainties introduced by the choice of calculation method of the $b$-coefficient (see section 2) were not considered in error propagation, but $b$-values determined by different methods were compared to assess their quality (see section 5.1).

Further systematic errors in REA-fluxes can arise from insufficient fulfillment of the prerequisites and assumptions, which underly all micrometeorological flux measurement methods (see e.g. Baldocchi et al., 1988). Therefore, results of field campaigns are often assessed by evaluating turbulence development (see supplement section S7 for the field campaign in Melpitz). Furthermore, the correct determination of the time lag between the change of the sign of the vertical wind and the actual switching of the valves is vitally important for the attribution of the sampled air to the two measurement channels. Imprecise time lags can cause significant errors on determined fluxes, especially in REA systems with long inlet lines (Moravek et al., 2013). The time lags of our system depend mainly on the sample flow rates and were quite short caused by the short inlet line and the high gas flow rate. Due to the different air flow rates in the inlet, different time lags when switching from ambient air to deadband (57 ms) or from deadband to ambient air (123 ms) were individually considered (for details see section 3.3). Therefore, errors of the calculated fluxes caused by imprecise time lags are expected to be negligible for the present application.



## 5    Field campaign in Melpitz

### 5.1    General observations

The field campaign took place during the period 21-09 to 02-10-2020. During the first three days of the campaign the weather
was warm and sunny followed by a longer rainy period. On the evening of the third day, the LOPAP instrument was calibrated,
for which accidentally a three order of magnitude higher nitrite standard was used. Since the background absorbance was
significantly increased hereafter, the measurements were stopped and the long path absorption tubes were cleaned with NaOH.
After a new calibration with the required nitrite concentration of 0.01 mg $l^{-1}$ the flux measurements were started again after
the rainy period on the 27-09-2020 up to the end of the campaign (02-10-2020). Caused by the potential variation of the
sensitivity of the LOPAP instrument by the cleaning procedure and the missing calibration for the measurements before the
cleaning, the HONO data from the first three days are affected by potentially higher uncertainties and were not further
considered here.

Caused by the requirements of the REA-method (see section 2), 30 min averages of the data of all instruments were used for
further data evaluation. In addition, only those periods were used for which data from all instruments were available. Besides
the missing LOPAP data for the first week (see above), there was also a short failure of the data logger of the spectroradiometer
on the 29-09-2020. In addition, during calibrations of the $NO_x$ instrument and during zero and parallel ambient measurements
of the LOPAP also no flux data were available. Finally, data with wind directions from north-north-east (340-50°) were also
not considered, in order to avoid potential turbulence disturbance of the field rack.

During the last five days of the campaign the weather was dry and sunny again, with completely clear sky on the 01-10-2020.
The temperatures varied from 3.5 °C during night up to 20 °C during early afternoon. The relative humidity was in the range
50-60 % during early afternoon and increased to 100 % at late night/early morning (see Figure S6). The turbulence was well
developed between 8:30 and 16:30 h, while low wind speeds and stable stratification of the atmosphere caused higher relative
errors of the HONO fluxes in the evenings and during night-time (see supplement section S7).

The $NO_x$ concentrations were relatively high for this rural measurement site. NO decreased from up to 12 ppb in the morning
to the detection limit of the instrument (30 ppt) in the late afternoon. $NO_2$ showed smaller variability with mixing ratios in the
range 2-10 ppb (see Figure S6). HONO levels varied in the range 2-280 ppt (see Figure 5), but did not show a typical urban
diurnal variation with high concentrations during night-time decreasing after sunrise to a minimum in the late afternoon (see
for example Figure 3 at the University of Wuppertal). Especially for the 30-09 to 02-10-2020 in Melpitz the HONO
concentrations were decreasing during the night and then increased again in the morning with a maximum before noon. Since
the $NO_x$ levels were also high at that time (see Figure S6), the late morning peaks may be explained by delayed arrival of
polluted air masses, e.g. from the morning rush hour in Leipzig.

Interferences were determined both, for the updrafts and downdrafts by regular changing of the allocation of the air masses to
the physical channels of the instrument. Mean values of $interference_{up} = (7.0 \pm 5.5)$ % and $interference_{down} = (8.0 \pm 7.8)$ % were
determined, which were similar in between their combined variability. In addition, similar concentration dependencies of the
interferences were observed, which show decreasing interferences with increasing HONO levels, similar to other LOPAP
measurements (see Figure 5 in Kleffmann and Wiesen, 2008). Thus, the applied interference correction, using the interference
data only from one channel for both air masses is considered accurate enough to determine HONO fluxes by the three-channel
REA-LOPAP system.

The comparison of three methods for calculation of the *b*-values (see section 2, equations (4) - (6)) during the field campaign
in Melpitz using two *K*-values of 0.9 and 0.6 (see equation (2)) showed very good agreement, when data from 08:30 to 16:30 h
were used. The average daytime $b_w$ (equation (5)) was $0.35 \pm 0.02$ for K = 0.9 and $0.43 \pm 0.03$ for K = 0.6. These values are in
good agreement with daytime values of $b_{Proxy}$ determined with the sonic temperature as proxy scalar (see equation (4)) of 0.34





± 0.06 and 0.38 ± 0.05, respectively. Finally, when a typical experimentally determined value of $b_0 = 0.56$ (Katul et al., 1996; Baker et al., 1992) was used in equation (6), the parameterized $b_{model}$ were 0.36 and 0.39, respectively.

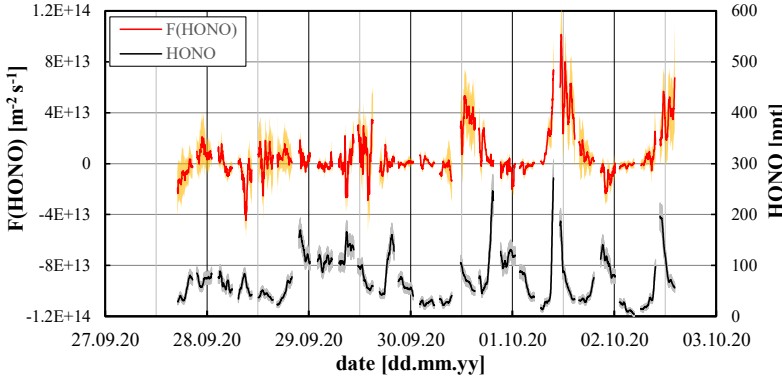

**Figure 5: HONO mixing ratios and HONO fluxes (30 min running mean data) in Melpitz. The grey and orange shaded areas reflect the accuracy of the data.**

### 5.2 HONO fluxes

The HONO fluxes showed a clear diurnal trend with low fluxes during night-time and a maximum around noon. The positive daytime fluxes were increasing to the end of the campaign (see Figure 5), which is explained by the previous strong rain period and the slowly drying soil surfaces during the following dry period. Since HONO is well soluble in water – particularly when in contact to slightly alkaline soil surfaces – high HONO emissions were initially prevented by the wet surfaces and increased only when the soil surfaces dried. During the last days a clear diurnal flux profile was observed showing low, partially negative fluxes (deposition) during night-time and higher positive fluxes (emission) around noon. During the second week of the campaign, the HONO fluxes varied in the range $-4 \cdot 10^{13}$ to $+1.0 \cdot 10^{14}$ molecules m$^{-2}$ s$^{-1}$ (see Figure 5).

For a better description of the general trends, a mean diurnal profile was calculated from the 30 min mean data of all instruments for the last four days, which is shown in Figure 6. As already described above for the single days, low and partially negative fluxes were observed during night-time, which were increasing to ~$4 \cdot 10^{13}$ molecules m$^{-2}$ s$^{-1}$ during daytime around noon.

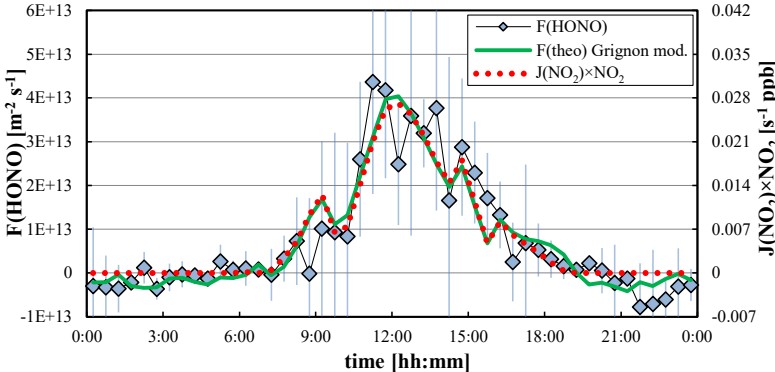

**Figure 6: Average diurnal day of the HONO fluxes of the last four days of the campaign in comparison to a modified parameterization from former gradient measurements in Grignon/France (Laufs et al., 2017), see section 5.4. Error bars represent the standard deviation of the HONO fluxes of the four days. In addition, the scaled product $J(NO_2) \cdot [NO_2]$ is also shown.**

While very similar diurnal trends were observed in other studies (Ren et al., 2011; Zhou et al., 2011; Zhang et al., 2012; Laufs et al., 2017) the average maximum HONO fluxes were lower than in these studies, where daytime fluxes of typically ~$1 \cdot 10^{14}$ molecules m$^{-2}$ s$^{-1}$ were observed. Only during the BEARPEX 2009 campaign (Ren et al., 2011) and in a forest clearing (Sörgel





et al. 2015) lower daytime HONO fluxes were observed compared to the present study. Reasons for the low HONO fluxes in Melpitz may be the solubility of HONO on the wet surfaces, as mentioned above, or lower nitrogen content in the non-fertilized soils of the Melpitz grass land compared to the other studies. In recent flux studies using twin open top soil chambers in China, up to two orders of magnitude higher HONO fluxes were observed shortly after strong fertilization of the soil surfaces (Tang 465 et al., 2019; Xue et al., 2019).

### 5.3 Potential HONO formation mechanisms

To identify potential HONO formation mechanisms, the diurnal averaged HONO fluxes were plotted against different potential precursors and parameters. The best correlation was observed between the HONO flux and the product $J(NO_2)\cdot[NO_2]$ ($R^2$ = 0.86, see also Figure 6), which is in excellent agreement with REA flux measurements during the CalNex 2010 campaign (Ren 470 et al., 2011) and gradient measurements in Grignon/France (Laufs et al., 2017). The results are also in good qualitative agreement with field studies, in which the daytime source of HONO was determined by the photo stationary state (PSS) approach. In these studies, good correlations of the HONO source were observed with radiation ($J(NO_2)$ or *irradiance*) and/or the product of *radiation·[NO_2]* (Vogel et al., 2003; Elshorbany et al., 2009; Sörgel et al., 2011; Villena et al., 2011; Wong et al., 2012; Lee et al., 2016; Meusel et al., 2016). These results point to a HONO formation mechanism identified in the 475 laboratory by the photosensitized conversion of $NO_2$ on organic surfaces, e.g. on humic acids (George et al., 2005; Stemmler et al., 2006; 2007; Bartels-Rausch, 2010; Han et al., 2016a; 2016b; 2017; Yang et al., 2021a).

Based on laboratory and field studies, other HONO sources are also discussed, for example the photolysis of nitrate/$HNO_3$ adsorbed on surfaces (Zhou et al., 2003; 2011), which requires shorter UV wavelength compared to the photosensitized conversion of $NO_2$ (see above). Unfortunately, nitrate concentrations on the soil and leaf surfaces were not measured in the 480 present study. However, it is expected that the maximum of this HONO source typically appears in the afternoon, since $HNO_3$ mainly forms by the oxidation of $NO_2$ by OH radicals in the gas phase during daytime, followed by the dry deposition of gaseous $HNO_3$ to ground surfaces. The diurnal maximum of gas phase $HNO_3$ in the afternoon was confirmed for the present field campaign (see Figure 7) and an even later maximum is expected for deposited $HNO_3$ on ground surfaces. Since the highest HONO fluxes were observed around noon (Figure 7), main HONO formation by nitrate/$HNO_3$ photolysis is unlikely 485 for the present field site. In addition, the correlation of the HONO flux with the product $J(O^1D)\cdot[HNO_3]$ ($R^2$ = 0.60) was weaker compared to the product $J(NO_2)\cdot[NO_2]$ ($R^2$ = 0.86). Here $J(O^1D)$ was used for the nitrate/$HNO_3$ photolysis because of the expected similar spectral ranges of the action spectra of $HNO_3$ and $O_3$ in the atmosphere (290 – 330 nm). Finally, only low photolysis frequencies $J(HNO_3)$ of adsorbed nitrate/$HNO_3$ were observed in recent laboratory studies (Laufs et al., 2017; Shi et al., 2021), confirmed by a field study, for which the nitrate photolysis was also excluded as a main HONO source in the 490 atmosphere (Romer et al., 2018).

The microbiological formation of nitrite in soils was proposed in laboratory studies as a further HONO daytime source (Su et al., 2011, Oswald et al., 2013). For this HONO source a strong negative humidity dependence was observed, which was explained by the solubility of HONO in soil water. In addition, the temperature of the upper layer of the soil was identified as an important parameter, for which the HONO emissions should increase with the soil temperature. Unfortunately, the soil 495 water content and temperature were not measured in the present field campaign. Due to the heat capacity of the soil, it is expected that the diurnal profiles of both parameters follow those of the air temperature and humidity. Since the highest air temperature and lowest relative humidity were observed at 15:00-17:00 h (see Figure 7), the maximum of a potential biological soil source is expected at similar or even later time of the day. In contrast, the highest fluxes were observed around noon near the maximum of the solar radiation (see Figure 6). In addition, low correlations of the HONO fluxes were observed with the 500 inverse of the relative humidity at 1 m height ($R^2$ = 0.26), with the difference (*100 % - RH*) ($R^2$ = 0.31) and with the near ground (1 m) air temperature ($R^2$ = 0.46), again contradicting a main HONO source by microbiological formation of nitrite in





the soil. In a field study by the group, which originally proposed the biological source, also a strong correlation of the HONO emissions with radiation was observed and, therefore, could not confirm their former laboratory results (Oswald et al., 2015). A further proposed daytime HONO source is the replacement of night-time adsorbed HONO/nitrite by strong acids on soil

surfaces, for example after deposition of $HNO_3$ (Donaldson et al., 2014; VandenBoer et al., 2014; 2015). However, also this mechanism seems to be not very plausible in the present study because of the main formation mechanism of $HNO_3$ and its diurnal mixing ratio profile, with a maximum in the late afternoon (see Figure 7). Caused by the subsequent dry deposition of $HNO_3$, an even later maximum is expected for the HONO source by acid replacement, in contrast to the observed HONO flux profile.

In a very recent study at the same measurement site in Melpitz, HONO emissions by evaporation of nitrite containing dew water was proposed (Ren et al., 2020). The nitrite was explained by the dry deposition and solubility of HONO in dew water during the preceding night, but which could not be confirmed by parallel dew and gas phase measurements. At first glance, this mechanism could be also plausible for the present field study, since the increase of the HONO concentration in the late morning on the 30-09 to 02-10-2020 (see above) appeared at the same time, when the relative humidity decreased from

saturation values (100 % RH), indicating the start time of dew evaporation. At the same time also the average HONO fluxes started to increase (see Figure 7). Dew measurements were not available for the present field study to confirm the proposed mechanism, which is however ranked unlikely here, caused by the smaller integrated deposition of HONO during the night compared to the positive daytime fluxes (see Figure 6). Integrated night-time deposition of HONO of only $5.9 \cdot 10^{16}$ molecules $m^{-2}$ was determined for the average day, much smaller than the integrated average daytime HONO emissions of $7.0 \cdot 10^{17}$

molecules $m^{-2}$. Thus, even when assuming an unreasonable quantitative re-emission of night-time deposited HONO, only at maximum ~8 % of the observed daytime fluxes could be explained by the proposed dew evaporation mechanism, which is thus considered unlikely for the present field campaign. The same quantitative argument is another hint to also exclude the above-mentioned acid replacement mechanism.

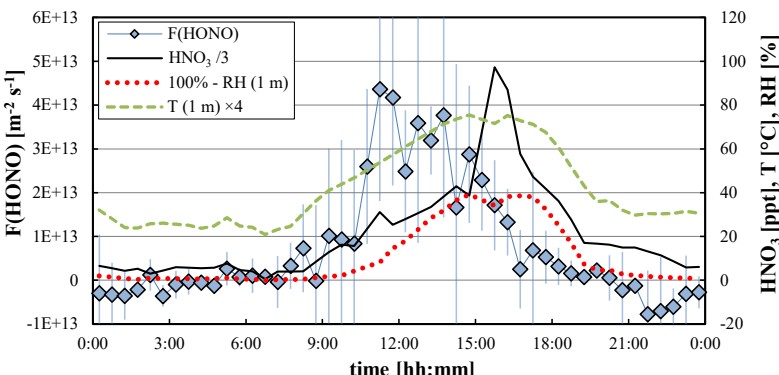

**Figure 7: Average diurnal profiles of the HONO flux, $HNO_3$ mixing ratio and the relative humidity (100%-RH) and temperature of the air in 1 m height. Error bars represent the standard deviation of the HONO fluxes of the four days.**

The temporal coincidence of the increasing HONO fluxes with the decreasing relative humidity in the morning and the high correlation with the product $J(NO_2) \cdot [NO_2]$ (see Figure 6)) imply HONO formation by photosensitized $NO_2$ conversion on ground surfaces, but only when the upper most surface layers get dry after the relative humidity is decreasing below saturation

values.

Finally, it has to be highlighted, that most of the discussed processes are still speculative due to the missing measured parameters (adsorbed $HNO_3$/nitrate on soil and grass leaf surfaces, humic acid and nitrite concentrations of the soil, soil humidity and temperature, strong acid deposition, etc.). Thus, the discussed sources could not be quantitatively described here.





Main focus of this field campaign were first flux measurements with the new REA-LOPAP system and not the final
identification of the most important source and sink processes, which is planned for the future.

### 5.4 Parameterization of the HONO fluxes

Due to the excellent agreement of the present study with results from former gradient measurements in Grignon/France (Laufs
et al., 2017), for which also the highest correlations of the HONO fluxes with the product $J(NO_2)\cdot[NO_2]$ and low nigh-time
deposition were observed, similar sources and sinks can be proposed for both rural field sites. In our former study the HONO
fluxes were well described by the following parameterization (Laufs et al., 2017):

$$(7) \qquad F(HONO)_{mod} = \left[\left(A \cdot J(NO_2) \cdot c(NO_2) + B \cdot c(NO_2)\right) \cdot exp\left(\frac{\Delta_{sol}H}{R \cdot T_{soil}}\right) - c(HONO) \cdot v(HONO)_T\right] \cdot \frac{RH}{50\%}.$$

The first and most important term in this equation ($A \cdot J(NO_2) \cdot c(NO_2)$) describes the photosensitized conversion of $NO_2$ during
daytime (see above). The second term ($B \cdot c(NO_2)$) is used to parameterize the night-time formation of HONO by heterogeneous
conversion of $NO_2$ on ground surfaces (Arens et al., 2002; Finlayson-Pitts et al., 2003). Because of the solubility of HONO in
soil water, both sources were scaled with the temperature by its enthalpy of solvation ($\Delta_{sol}H$). Since the emission of HONO
from soil surfaces and not its solubility in water is considered, a positive sign was used for this Arrhenius-term. The next term
($c(HONO)\cdot v(HONO)_T$) considers the temperature dependent deposition of HONO on the ground, which was observed to
increase with decreasing temperature (Laufs et al., 2017). Since it can be expected that both, sources and sinks scale with the
humidity (Finlayson-Pitts et al., 2003; Stemmler et al., 2006; Han et al., 2016; Su et al., 2011) all terms were finally
parameterized by the relative humidity ($RH/50\%$).

When using the original parameters from the former study in Grignon (Laufs et al., 2017), but the air temperature (1 m) instead
of the soil temperature, a factor of about three higher fluxes were calculated compared to the measurements in Melpitz, which
can be explained by the wet surfaces in the present field campaign (see above). To consider the differences between both
campaigns, the constants $A$ and $B$ were decreased to $A = 8.807\cdot10^5$ m and $B = 1118.4$ m s$^{-1}$. Other values in equation (7) (see
Laufs et al., 2017) were not changed. With this modified parameterization an excellent qualitative and quantitative agreement
with the measured fluxes was observed (see Figure 6), indicating that the main source and sink processes identified in Grignon
are also active in Melpitz.

### 6 Conclusion

A Relaxed Eddy Accumulation (REA) system for the quantification of vertical fluxes of nitrous acid (HONO) was developed
and tested, which is based on the LOPAP technique. Two fast acting valves are controlled by the sign of the vertical wind
speed and feed the sampled gas flow from the updrafts and downdrafts to two LOPAP channels. The developed PyREA
software allows controlling measurement cycles, which regularly alternate between REA-, zero- and parallel ambient
measurements. Only small differences of the interferences were identified for the updrafts and downdrafts, excluding
significant errors when using only one interference channel. In laboratory experiments, high precision of the two channels and
the independence of the dilution corrected HONO concentrations on the length of the valve switching periods were
demonstrated.

A field campaign at the TROPOS monitoring station in Melpitz, Germany showed campaign averaged diurnal HONO fluxes
in the range of $-8\cdot10^{12}$ molecules m$^{-2}$ s$^{-1}$ (deposition) during night-time increasing to $+4.4\cdot10^{13}$ molecules m$^{-2}$ s$^{-1}$ (emission)
around noon. The positive HONO emissions during daytime were continuously increasing after an intensive rainy period,
which was explained by the drying of the upper most ground surfaces. Similar to other campaigns the highest correlation of
the HONO flux was observed with the product $J(NO_2)\cdot[NO_2]$, which implies a HONO formation by photosensitized conversion
of $NO_2$ on organic surfaces, like e.g. humic acids.





## 7 Acknowledgement

The Deutsche Forschungsgemeinschaft (DFG, German Research Foundation) is acknowledged for the financial support under
the contract number KL 1392/4-1. We also would like to thank the TROPOS institute and especially Hartmut Herrmann,
Gerald Spindler, Achim Grüner and Laurent Poulain for enabling the field campaign at the measurement station in Melpitz,
for their continuous and very helpful support during the campaign and for providing the meteorological data and the data for
ozone and nitric acid. And finally, we would like to thank Andreas Held for fruitful discussions and for his feedback and advice
regarding the REA system.

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
