# Peer review of "A Relaxed Eddy Accumulation (REA) LOPAP-System for Flux Measurements of Nitrous Acid (HONO)"

_Atmospheric Measurement Techniques, 2021_

## Author Comment (AC1)

**Reply to Referee #1**

**General comments:**

The authors of this manuscript set up a REA system for measuring the vertical fluxes of HONO based on development of a three-channel-LOPAP instrument for HONO measurements of the updraft, downdraft and chemical interferences as well as a software for controlling valves and measurement cycles. The system was well tested in both laboratory and field to be reliable for the measurements of HONO vertical fluxes. In general, this is a nice research work, which can direct researches to conduct field measurements of HONO vertical fluxes to comprehensively understand the atmospheric HONO sources in different areas.

We would like to the thank the referee for her/his interest in our study and the detailed comments, which will help to improve our manuscript.

**Specific comments:**

**Are the fluxes measured through the REA method affected by the height above the ground? Which is the height proper for selection?**

Commonly, it is assumed that the flux in the surface layer is independent of the height above the ground. However, the accuracy of the measurements will certainly depend on the height, since the size distribution of the turbulences (Eddies) is height-dependent and their size will on average decrease towards the ground. If the Eddies are too small and their frequency too high, they cannot be resolved by the REA method, due to the size of the anemometer and the used sampling frequency (10 Hz). Foken (2008) recommended a measurement height of at least 2 m for a measurement path length of 12 cm, which is similar to the path length of the sonic anemometer that was used in this study (11.6 cm). Thus, the height above the ground should not be too low. In addition, if HONO fluxes are measured high above the surfaces, where HONO is expected to be formed, e.g. at 20 m above the ground, the fast photolysis during the transport from the source region to the instrument will cause an underestimation of the daytime flux. Thus, in the case of HONO, the measurement height should also not be too high. For measurement heights in the range 1.5-3 m above canopy surfaces, similar to that used in the present study, photolytic losses during the transport were estimated to be <10 % in Laufs et al. (2017). Furthermore, the footprint of the measured fluxes is dependent on the measurement height. To obtain fluxes that represent a net signal of the near surroundings, the measurement height should not be too high, as well. Finally, the chosen height of 2 m was also for practical reasons, because the sampling unit had to be accessible for the calibration procedure.

Unfortunately, we have however still not systematically studied the impact of the measurement height on the quality of the flux measurements for our REA system.

There are several prerequisites and assumptions for the REA method. How to verify the reliability of fluxes measured by the REA method?

Since flux errors are caused by insufficient fulfilment of the underlying assumptions, a careful quality assessment is important to judge the reliability of the measured fluxes. The most important prerequisites for reliable micrometeorological flux measurements and also the REA

method are well developed turbulence, homogeneity of the surface and stationarity (steady state) of the process.

We examined the state of turbulence as described in section S7 of the supplement to select periods during the day with good conditions for flux measurements. The turbulence can be described as well-developed only between 8:30 h and 16:30 h because of rather weak winds during most of the days of the campaign (see section S7). This interval meets the time of positive HONO fluxes and cover the daytime period, which is of main interest for the discussion of potential HONO sources in section 5.3. Night-time fluxes, on the contrary, are less reliable because of low turbulence (lines 416-418).

The horizontal homogeneity of the surface, which avoids advective effects disturbing the fluxes, is sufficiently ensured by the flat terrain without obstacles in the surroundings of the measurement site (see description of the site in section 3.5). In the revised manuscript, we will add the sentence "The terrain is uniform and even, which ensures horizontal homogeneity of the flow field and makes the site well suited for micrometeorological measurements." in line 253 to make this point clearer.

We also performed the stationarity test after Foken and Wichura (1996) to check the steadystate assumption. But as the amount of available data was already sparse, we decided not to discard any data due to insufficient stationarity. Most of the 30-min periods that were tagged as instationary were during night-time and in the early morning. The fluxes during these times were rated as more uncertain because of low turbulence anyway (s. above). The daytime fluxes were mostly of good quality with regard to this test. However, a detailed discussion of the results of the stationarity test would have exceeded the scope of this more technical paper.

Finally, in the future, the reliability of our REA-system should be verified by comparison with other flux systems, e.g. the gradient method or open flux chambers, which however, will also have their specific uncertainties. Up to now, we did not have the opportunity for such an intercomparison. Based on the well-developed turbulences during daytime, we think that our daytime fluxes are sufficiently reliable for the analysis of potential HONO formation mechanisms.

**How did you measure the sampling efficiency of 99.6% for HONO? Part of HONO signal from the Channel 2 might be attributed to breakthrough from the Channel 1, rather than the chemical interference.**

The sampling efficiency of stripping coil 1 was measured by a pure HONO source (Villena and Kleffmann, 2021) from the ratio of the signals in channel 1 and 2 (double stripping coil) for different air flow rates. The logarithm of the loss of HONO from channel 1 was then plotted against the inverse of the flow rate (proportional to the residence time in the coil). The excellent linear correlation confirms the first order uptake kinetics in the coil. The regression fit can be used to calculate the sampling efficiency for any flow rate and a sampling efficiency of 0.996 was calculated for the experimental conditions of the present study. This is in excellent agreement with the experimental sampling efficiency of  $0.9965\pm0.0029$  obtained for the data shown in the revised Figure 4 (see below). Since the geometry of all three coils was made as similar as possible by our in-house glass blower, we assumed the same sampling efficiency for all coils. Since the sampling efficiency is close to 1, even any small differences of the coils will not significantly affect the results. E.g. if coil 3 would have a 10 % smaller surface area

compared to coil 1 (this is by far the upper limit), than the sampling efficiency would decrease only from 0.996 to 0.993 %. Even in this case, the difference of the resulting HONO concentration would be smaller than the precision error of the instrument.

To clarify, we will add in section 3.1 behind the specified sampling efficiency: "The sampling efficiency was determined by using a pure HONO source (Villena and Kleffmann, 2021)."

To the second part of the comment: The small HONO losses from coil 1 and 3 are considered in the data evaluation. First, the signals of coil 1 and 3 are divided by 0.996 to account for the loss of HONO by incomplete sampling. Second, the loss of HONO to the interference channel 2 (channel 1.0.004) is subtracted from channel 2 and only the remaining interference is subtracted from the signals in channel 1 and 3 (see lines 317-319).

Were the wind directions affected by the small cone at the inlet? Small turbulence of wind could occur around the cone, which may affect the vertical fluxes.

Since the inlet of the LOPAP including the small cone was positioned leeward below the lower arm of the anemometer (see lines 167-168) and since the diameter of the cone (3 cm) was small compared to the vertical distance between the inlet and the midpoint of the sonic transducers (~30 cm), turbulences induced by the cone should not significantly affect the turbulence measurements. We followed here the recommendations of Kristensen et al. (1997), but did not systematically study the influence of the inlet position on the turbulence measurements.

**Technical corrections:**

**Line 292, I don't know why the time correction is about 3 times of the time response.**

In a LOPAP instrument the signal always arrives delayed, since the liquid reagents from the stripping coil are transferred to the instrument through the 3 m long reagent line and react with reagent 2 before arriving to the long path absorption tubes of the instrument, where the dye is detected. The time correction is defined by the time between any incident (e.g. start of a zero measurement) until the signal reaches the 50 % change to the final value. This time correction was 16 min under the experimental conditions applied in Melpitz and by this time the measured signals are "shifted back" to get the correct concentration time profiles. The time correction of the LOPAP technique was often verified, e.g. in smog chambers by the comparison with other fast instruments.

Besides this LOPAP-specific correction, any instrument has a physical response time, which is defined here as the time from the first signal change until 90 % of the final signal is reached. This response time was 5 min for the experimental conditions applied, and is mainly caused by the liquid flow rate, by the exchange times of the reagents in the glass Ts (for mixing, de-gassing etc., see Figure S2) and the length of the AF2400 long path absorption tube. To get the same averaging interval for the LOPAP and the turbulence data, this time is chosen also as ringbuffer time W1 (see Figure S3) in the PyREA software.

Both specified times (time correction and response time) are depending on the liquid flow rate, but their ratio of ca. 3 is accidental and would be e.g. different if a shorter reagent line or a different absorption path length were used.

Line 293, ",." should be ",".

Will be corrected in the revised manuscript.

*Line 301, "P" should be "P".*

Will be corrected in the revised manuscript.

**Line 315-316, I don't understand the meaning of this sentence. Why were the channels 1 and 2 divided by the half of the ratio, whereas channel 3 multiplied the ratio?**

There is a small difference between the HONO concentrations in both channels determined during the parallel ambient measurements caused by small temporal drifts of the sensitivities of both channels compared to those determined during the calibration. Typically, these differences are of the order of 1-2 %, similar to the precision error of the LOPAP. However, even such small difference would result in a 10-20 % error of the HONO flux, if the concentrations in the two channels differ only by 10 % during REA measurement. To minimize the flux errors, both channels were harmonized by using the time dependent ratio of channel 1 / channel 3 during regular parallel ambient measurements. Since we do not know, which of the two channels measures correctly, we harmonized to the average of both, i.e. the smaller signal of one channel was increased and the larger signal of the other channel decreased by half of the difference.

We will rephase in the revised manuscript: "Since it is unclear, which of the channels measures correctly, all channels were corrected only by half of the ratio (channel 1 / channel 3) determined during the parallel ambient measurements, for which channels 1 and 2 were divided and channel 3 multiplied by this ratio, respectively."

Line 329, the air flow rate of 3.7 l/min in here was inconsistent with that of 2.65 l/min mentioned in line 180.

In line 329 we discuss potential heterogenous HONO formation in the PFA inlet line during REA measurements, for which the total flow rate in the inlet line has to be considered (3.7 l min-1), which is the sum of the bypass flow (2.65 l min-1, see line 180) and the sample flow of one channel (1.05 l min-1, see line 127). During REA measurements, only one of the two channels is sampling through the inlet, while the other channel is under zero air.

In the revised manuscript, we will add after the total flow rate: "(sum of bypass and sample flow rates of 2.65  $l \min^{-1}$  and 1.05  $l \min^{-1}$ , respectively)".

Line 365, the dilution seemed to vary with time, e.g., the dilution between 13:00 and 14:00 was less than a factor of 4, which may result in significant uncertainty of fluxes.

The experiment presented is unfortunately not ideal for this dilution test, since the HONO concentration still varied in the lab. Thus, the dilution corrected HONO concentrations still showed temporal variations, see the following figure (similar to Figure 4, however, corrected for dilution, the short REA period was not corrected):

Caused by the temporal variations of the HONO concentrations, we specified only the average ratio between all parallel ambient measurements and all valve switching periods, for which we confirmed the theoretical dilution by a factor of four. However, due to these non-ideal conditions and the reasonable concern by the reviewer, we have repeated the experiment shown in Figure 4 by using a pure HONO source (see Villena and Kleffmann, 2021) at stable HONO concentrations, which will be presented in the revised manuscript. Here the factor of four dilution could be verified for all steps and not only for the average.

Figure 1: HONO mixing ratios during test measurements using a pure HONO source and variable valve switching times (30 s data). "Z": zero; "P": parallel ambient measurements; "xs": valve switching experiments with constant periods for up-, down and dead-band measurements (each with 1, 2, 3, 4 s using the sequence: up, dead-band, down, dead-band; dilution each by a factor of four).

We will also adopt the text in section 4.2 accordingly:

"After correctly adjusting the pressure in the REA inlet (see section 3.2) the valve switching periods were systematically varied by using the PyREA test-mode *valve\_function\_test* (see

section 3.3) at constant HONO concentrations by using a pure HONO source (Villena and Kleffmann, 2021). A sequence of zero- (both valves closed), parallel ambient measurements (both valves open) and subsequent valve switching periods was performed, for which constant valve switching times of 1, 2, 3 and 4 s were applied for the periodically switching valve positions (1) up, (2) deadband, (3) down and (4) deadband, respectively. Caused by the chosen valve switching sequence, the HONO concentration is diluted by a factor of four in both channels during these experiments (see Figure 4: "P" and "xs").

For the parallel ambient measurements an excellent agreement between channels 1 and 3 was again observed, with an average ratio (1/3) of  $1.002\pm0.006$ . Thus, no further harmonization of the two channels (see section 4.1) was performed for the short measurement period applied. For the valve switching test the expected dilution by a factor of four could be confirmed in both channels. Here, a mean ratio (channel 1 / channel 3) for the dilution corrected data of  $1.023\pm0.0012$  was observed. The small deviation of 2.3 % is in between the precision error of the instrument, considering also the applied dilution correction and is independent of the chosen switching time interval. In addition, no significant differences between the HONO data during parallel ambient measurements and the dilution-corrected data during the valve-switching experiments were observed. For the average ratios (parallel ambient measurements / valve switching) values of 0.984 $\pm$ 0.015 and 1.005 $\pm$ 0.017 were observed for channels 1 and 3, respectively. Thus first, the agreement between both channels is independent of the valve switching period and second, the dilution is correctly determined by the system, which is the basic requirement for REA-measurements."

Lines 368-374, I don't know why the different average ratios of channel 1 / channel 3 were present at different places for the same experiment despite of small difference. Line 369, "ratio (1/3)" should be "ratio (channel 1 / channel 3)".

As recommended, "ratio (1/3)" will be replaced by "ratio (channel 1 / channel 3)" in the revised manuscript.

Furthermore, the ratio is presented several times to confirm:

a) the good agreement during the parallel ambient measurements to exclude any further harmonization (see lines 368-370) as done for the longer campaign data in Melpitz;

b) the dilution was similar in both channels during the different valve switching periods (independent of the absolute dilution factor), since the same concentrations were obtained in both channels (see line 371) and

c) the average dilution for all valve switching periods was indeed four for both channels (with some temporal variations in the laboratory, see answer to the last comment), since the average concentrations during the parallel ambient measurements were similar to the average dilution corrected concentrations during the valve switching periods (corrected by the nominal dilution factor of four) in both channels (see lines 371-374).

*Lines 400-402, the detail information for maloperation of the instrument is not necessary.*

In this section, we tried to explain, why only data from the last five days of the 11 days campaign could be used for evaluation. As recommended, we will focus the description only on the last

week of the campaign (27.09. - 02.10) and will delete all information of the first part throughout the manuscript to simplify the discussion in the revised manuscript.

**Lines 429-431, Fig.5 seemed not support the conclusion.**

Here the referee has overlooked that the reference in line 431 not refers to Figure 5 of the present study, but to Figure 5 of our former paper Kleffmann and Wiesen (2008). In lines 429-431, we describe that the relative interferences for the updraft and downdraft air masses were not only similar on average, but also showed a similar concentration dependence. Here, we always see increasing relative interferences with decreasing concentration with our LOPAP instruments. Since this very detailed interference data is out of the scope of the present study and thus not shown here, we simply referred to the former study, which was specifically aimed to describe the interference issue of wet chemical instruments, like the LOPAP technique.

*Lines 431-433, the sentence is suggested to be moved before the sentence of "In addition..." for logical connection.*

No, the logical order is correct. Both, the similarity of the average relative interferences for both air masses, and their similar concentration dependencies show that the interference correction applied in the present study is justified, for which only one interference channel is used for both air masses.

**Lines 459-460, this sentence is suggested to be rephrased due to unclear description.**

We will rephrase to: "In other studies very similar diurnal trends of the HONO flux were observed, which are following the diurnal trend of the radiation (Ren et al., 2011; Zhou et al., 2011; Zhang et al., 2012; Laufs et al., 2017). However, the average maximum HONO fluxes of typically  $\sim 1.10^{14}$  molecules m-2 s-1 were higher by factors of 2-3 in these studies compared to the data shown in Figure 6."

Line 466, the reasons for exclusion of the HONO sources other than the photosensitized conversion of NO2 on organic surfaces are not very convincing, because the authors didn't consider the possible heterogeneous HONO formation in dew and ground surfaces during night, which may be high enough to explain the large difference between the negative flux at night and positive flux in daytime.

We generally agree with the referee, that the discussion about the potential HONO formation mechanisms given in section 5.3 is still speculative, since important parameters were not measured in the present study, which was aimed mainly to test our new REA-LOPAP system in a field campaign. We have already highlighted this shortcoming at the end of this section in lines 531-535.

Furthermore, the deposition of HONO may indeed underestimate the nitrite levels adsorbed on the ground or in the dew as mentioned by the referee, since nitrite may be also formed by the deposition and heterogeneous conversion of NO2 on humid surfaces – the main proposed night-time formation mechanism of HONO on ground surfaces. In contrast, any deposition of HONO formed in the gas phase or on particles, would be accounted for by our flux data. Other HONO/nitrite formation mechanisms on ground surfaces (e.g. HNO3 photolysis, biological production) will be absent or small during night-time. For the heterogeneous conversion of NO2 any simultaneous re-emission of HONO to the atmosphere can be excluded, since we observed neutral to small negative HONO fluxes at night. This is also in agreement with the decreasing HONO/NOx ratios, showing that night-time formation of HONO was not efficient in Melpitz.

Now to calculate the maximum nitrite levels accumulated at the end of the night on ground surfaces by heterogenous conversion of NO2, we can consider a reasonable dark reactive uptake coefficient of NO2 of 10-6 (Kurtenbach et al., 2001). For such a small uptake coefficient, transport limitations will not be significant and would only further decrease the accumulated nitrite. If we take the average NO2 concentration during night-time of 4.8 ppb in Melpitz and assuming a 12 h night, at maximum additional  $2.35 \cdot 10^{17}$  m-2 nitrite could accumulate on the ground surfaces assuming a HONO yield of 50 % (by 2 NO2+H2O $\rightarrow$ HONO+HNO3). Even if we then assume an unreasonable quantitative re-emission of the adsorbed nitrite during daytime, only a third of the observed positive daytime fluxes could be explained by the mechanism proposed by the referee. However, a quantitative re-emission of the HONO formed by the heterogeneous NO2 conversion was yet not observed. In the two field studies, which we are aware of by Stutz et al. (2002) and Laufs et al. (2017), only 3 % and 2-4% of the deposited NO2 was re-emitted as HONO, respectively. This may be explained by additional losses of nitrite on ground surfaces, e.g. by oxidation to nitrate or bacterial conversion. If a more reasonable fraction of re-emitted HONO of 4 % is considered, only 2.7 % of the measured positive HONO fluxes during daytime could be explained by this mechanism. Furthermore, even if one assumes any higher night-time NO2 uptake and a quantitative re-emission during daytime (both are unlikely), it is very reasonable that the daytime fluxes of HONO by this reemission mechanism would show a peak in the morning, when the ground surfaces are irradiated by sunlight leading to the evaporation of the dew. This was observed in all studies, where the dew evaporation mechanism was investigated (e.g. He et al., 2006; Ren et al., 2020). In contrast, a re-emission of the adsorbed nitrite, following exactly the diurnal shape of the product of radiation and NO2 concentration throughout the whole day (see Figure 6), is implausible. Also, if the acid replacement mechanism is considered (uptake of strong acids replacing the night-time accumulated nitrite as HONO to the gas phase) a different diurnal shape of the HONO flux is expected (see discussion in the manuscript).

Thus, although the discussion about the potential HONO formation mechanisms during daytime in Melpitz is indeed speculative, we are quite confident that the proposed photosensitized conversion of  $NO_2$  is still the most probable one at the field site. We will add the above discussion to the revised manuscript.

**References used in this reply and not listed in the manuscript:**

Foken, Th.: Micrometeorology, Springer, Berlin /Heidelberg, 2008.

Foken, Th. and Wichura, B.: Tools for quality assessment of surface-based flux measurements, Agric. For. Meteorol., 78 (1-2), 83-105, doi: 10.1016/0168-1923(95)02248-1, 1996.

He, Y., Zhou, X., Hou, J., Gao, H., and Bertman, S. B.: Importance of Dew in Controlling the Air-Surface Exchange of HONO in Rural Forested Environments, Geophys. Res. Lett., 33, L02813, doi: 10.1029/2005GL024348, 2006.

Kurtenbach, R., Becker, K. H., Gomes, J. A. G., Kleffmann, J., Lörzer, J. C., Spittler, M., Wiesen, P., Ackermann, R., Geyer, A., and Platt, U.: Investigations of Emissions and Heterogeneous Formation of HONO in a Road Traffic Tunnel, Atmos. Environ., 35, 3385-3394, doi: 10.1016/S1352-2310(01)00138-8, 2001.

Stutz, J., Alicke, B., and Neftel, A.: Nitrous Acid Formation in the Urban Atmosphere: Gradient Measurements of NO2 and HONO over Grass in Milan, Italy, J. Geophys. Res., 107 (D22), 8192, doi: 10.1029/2001JD000390, 2002.

Villena, G., and J. Kleffmann: A Source for the Continuous Generation of Pure and Quantifiable HONO Mixtures, Atmos. Meas. Technol. Discuss., doi: 10.5194/amt-2021-332, 2021.

---

## Author Comment (AC2)

**Reply to Referee #2**

*The actual mechanism for ground/surface source of HONO is still up for debate, and high-quality flux measurements will aid in answering this question. The work by von der Heyden et al, describes an adapted LOPAP system for determining HONO fluxes by REA. In my opinion, the authors have done an good job describing the instrument, calculations and have provided a number of lab and field tests to validate the instrument for measuring fluxes. I especially appreciated how the authors listed some of the problems encountered related to flow rates and inlet pressure between the different channels and the solutions in section 3.2. It was refreshing to see, as presenting this sort of information will help other solve similar problems in the future.*

We would like to thank referee #2 for her/his interest in our work and also the helpful comments, which are addressed below.

*I would have liked to see more characterization or validation on the effect of having an 80cm inlet before the stripping coil on the measured HONO. HONO formation on inlet surfaces is well known, as the authors state in the text, and this inlet in my opinion reduces one of the advantages of a normal LOPAP system (i.e. the very short inlet prior to sampling). While the authors attempted to mitigate it by covering it in foil, I would have liked to have seen some experimental validation that this inlet wont lead to a bias in the measured HONO, rather than just stating that the small residence time means it won't matter.*

We fully agree with the referee's concern, which we already discussed in the manuscript as a potential problem (see lines 168-172 and 327-332). However, unfortunately it is impossible for a REA-system to avoid the use of inlet surfaces, since a common inlet line, the two valves and their connections to the inlets of the two stripping coils are mandatory. In addition, caused by the dimension of the external sampling unit, there is a minimum distance to the anemometer necessary to avoid disturbances of the turbulence measurements, even if the sampling unit is placed in the lee of the anemometer. Due to these requirements we had to use a 80 cm long PFA inlet tube (4 mm i.d.) shielded against irradiation by aluminum foil to prevent for photochemical formation of HONO, e.g. by photolysis of nitrate on inlet surfaces. Since we are aware of potential inlet artefacts even in a dark inlet – that was the reason why we developed the external sampling unit for the LOPAP technique – we also increased the total inlet airflow rate to 3.7 L $min^{-1}$ by adding the bypass flow, minimizing the gas-surface interaction time. It should be highlighted that the length of our inlet is much shorter than those used in other REA studies further minimizing the inlet artefact.

From our experience with inlet tests during the development of the LOPAP technique, artificial formation of HONO was significant when using a 3 m long heated PFA inlet line (4 mm i.d.) and a flow rate of only 1 L $min^{-1}$, especially at low HONO/$NO_x$ ratios and low HONO levels during daytime. In contrast, during night-time at higher HONO/$NO_x$ ratios and higher HONO levels, the artificial HONO formation was of less importance. However, since the gas residence time in this former inlet was almost 14 times longer than in the 80 cm short inlet used for the present REA system (see 3 m vs 0.8 m and 1 L $min^{-1}$ vs 3.7 L $min^{-1}$), we do not expect significant heterogeneous HONO formation for our REA-LOPAP even at low HONO levels and low HONO/$NO_x$ ratios. In addition, since this heterogeneous artificial HONO formation would only affect the absolute HONO levels and not the concentration differences, it would not influence the HONO fluxes. Fluxes are calculated from the concentration difference of the up- and downdrafts (see equation (1)). So even if the absolute concentrations of both air masses are

affected by significant artificial HONO formation on inlet surfaces, that would affect both air masses to the same extent and would not change the concentration differences and fluxes.

However, since we also calculate average absolute HONO concentrations using the REA data (see Figure 3), for the next field campaign, we plan to intercompare the REA-instrument with a common LOPAP, for which no inlet surfaces are used. Unfortunately, we still do not have data from such an intercomparison.

*Overall, this manuscript is well written, clearly presented and will be of interest to many in the community.*

We would like to thank referee #2 for this positive statement.

*Minor Comments*

*Line 25: As you state in section 5.3, much of your discussion on potential HONO sources is speculative. I am not sure you can claim these HONO formation mechanism are minor.*

We not generally claim other formation mechanism as minor, but only for the present field site. Based on the correlation of the daytime fluxes with different parameters, other formation mechanisms than the proposed photosensitized conversion of $NO_2$ are less likely in Melpitz, see discussion in section 5.3. With respect to a similar comment by referee #1, who suggested an alternative explanation of our results, we will also extent the discussion in section 5.3 by additional quantitative calculations. However, also these calculations show that the alternative mechanism suggested by referee #1 is of minor importance (see our reply to referee #1). We would appreciate if referee #2 has some additional suggestions how other proposed mechanisms could better explain our field data, which could be included into the discussion section.

To consider the concern of referee #2, we will weaken the sentence in line 25 by: "…, but are tentatively ranked being of minor importance for the present field campaign."

*Line 140: there have been more recent instrument intercomparisons comparing HONO measurements from the LOPAP to other instruments (Crilley et al., 2019; Pinto et al., 2014), which indicate that there can be significant variability in the reported measured values of HONO between the LOPAP and other established techniques . This should be indicated here.*

We are aware also of several other intercomparison campaigns (e.g. Ródenas et al., 2013), where LOPAP instruments agreed not as well to other techniques as in our cited former study Kleffmann et al. (2006). However, we consider this cited study as more relevant for the present study for the following reasons:

First, in our former study, the LOPAP was intercompared with the DOAS technique for the same air mass, since a White mirror long path absorption system collocated to the LOPAP inlet was used for the DOAS (= both instruments in-situ). Thus, we had no problems with spatial inhomogeneous air masses discussed e.g. in the study of Crilley et al., 2019 and also for some

instruments in Pinto et al., 2014 (here in-situ instruments were compared with LP-DOAS and the inlet of the LOPAP was different to the co-located inlets of the MC-IC, SC-AP and QC-TILDAS).

Second, in Kleffmann et al. (2006), neither the LOPAP nor the DOAS used any significant surfaces on which artificial HONO could be formed (long inlet lines, cavity cells, etc.), in contrast to some instruments used in the two intercomparison studies mentioned by the referee.

Third, in Kleffmann et al. (2006) our LOPAP was operated by ourselves (similar to the present REA-LOPAP) using a similar basic set-up and data treatment, which may be not the case in other studies. For example, in the FIONA intercomparison campaign (Ródenas et al., 2013), several LOPAP instruments were intercompared, some of which showing significant differences. However, reason for that, at least in part, was the individual operation of the instruments and for two LOPAPs also modifications of the original set-up. We proposed this set-up and the data treatment several years ago during the instrument's development and still use it, e.g. for the present REA-LOPAP. Generally, any instrument can produce low quality data, if not well operated. Even the DOAS instrument, which is typically defined as the "gold standard", only worked properly in the EUPHORE chamber, after a negative artefact caused by impurities of HONO in the $NO_2$ reference spectra was considered for (see Kleffmann et al., 2006), which is a general problem of all spectroscopic instruments working in the near UV (see also IBBCEAS).

In conclusion, by the sentence in line 140 we would like to highlight that our LOPAP instrument operated by ourselves agreed well with the DOAS technique, which might not be the case for other LOPAP instruments operated by other users for several reasons (see above). Thus, we consider the cited intercomparison as more relevant for the present study. A detailed discussion about the various intercomparisons including different LOPAP instruments is however out of the scope of the present study.

*Figure 1: Why does stripping coil for channel 3 only have one coil? Wouldn't it be better if you also quantified the interferences in this coil as well?*

Yes, we agree! But that was technically not possible in a standard LOPAP housing and two complete standard LOPAPs could not be integrated into the field rack, which was already quite bulky for flux measurements (see supplement Figure S5). We have discussed this issue and the validation of the applied interference correction in detail in lines 142-149, 337-346 and 427-433.

*Line 311-319: I struggled to follow the explanation for the corrections applied for dilution during REA measurements. This may be better presented as equation. Furthermore, it wasn't clear to me channel 3 was treated differently to channel 1 and 2.*

For both running averaging intervals (W1 and W2, see supplement Figure S3) every 30 s the PyREA software records fractions of how long each channel is sampling ambient air, which could be, for example, a third of the total averaging interval. For the other two thirds of the time, when the instrument is in the dead band, or when the other channel is sampling ambient air, zero air is sucked into the stripping coil. Thus, the measured HONO concentration reflects

a diluted sample, with a dilution ratio defined by the valve switching statistics. However, since we need the undiluted HONO concentration for the flux calculation by equation (1), the dilution was corrected for. E.g., in the example above, the measured diluted concentration would be multiplied by a factor of three. For this dilution correction, we used the shorter running average interval W1 of 5 min for the valve switching statistics, which was adjusted before the campaigns in the PyREA software to the physical time response of the LOPAP instrument. Every measured data point of the LOPAP (30 s) also reflects a 5 min running average, which is caused by the LOPAP instrument's measurement principle. For the dilution correction all channels were treated the same way. Only for channels 1+2 the valve switching statistics of valve A and for channel 3 the valve switching statistics of valve B were used (see Figure 1).

Next, the undiluted data was harmonized based on the results of the parallel ambient measurements, caused by small drifts of the instrument's sensitivity compared to the calibration results. Here again all channels were treated by the same correction. Since we do not know, which of the two channels measures correctly, we harmonized to the average of both, i.e. the smaller signal of one channel was increased and the larger signal of the other channel decreased by half of the difference of the parallel ambient measurements. Channel 2 (interferences) was harmonized in the same way than channel 1, since both channels sample the same air mass (see Figure 1). However, caused by the typical small signals in channel 2 (ca. 10 % of channel 1), any errors in the harmonization of channel 2 (typically only a 1-2 % correction), will not significantly affect the accuracy of the HONO data (i.e. by only 0.1-0.2 %, much smaller than the precision error of the instrument). Also with respect to referee #1, we will explain the harmonization in more detail in the revised manuscript (see our reply to referee #1).

After harmonization, the HONO concentrations of the up- and downdrafts were calculated in the same way as done for a normal LOPAP instrument. Here, from channels 1 and 3 (measuring HONO + interference) the signal from channel 2 (measuring only the interferences) is subtracted and the incomplete sampling of HONO in channels 1 and 3 is considered. For the correction, the signals of channels 1 and 3 are divided by the sampling efficiency of 99.6 % (see line 127) to account for the loss of HONO by incomplete sampling. In addition, the loss of HONO to the interference channel 2 (channel 1·0.4 %) is subtracted from channel 2 and only the remaining interference is subtracted from the signals in channel 1 and 3 (see lines 317-319). Also this interference correction was done in the same way in channels 1 and 3, thus we do not understand the last concern of the referee?

Finally, we agree that the data treatment is quite complex and we will try to explain that more clearly in the revised manuscript.

*Line 319: what is HONO sampling efficiency of the stripping coils? I couldn't find that value in the text.*

The sampling efficiency is specified in line 127 (99.6 %) and was verified by a pure HONO source. See also similar question by referee #1.

References not used in the manuscript:

Ródenas, M., Muñoz, A., Alacreu, F., Brauers, T., Dorn, H.-P., Kleffmann, J., and Bloss, W. J.: Assessment of HONO Measurements: The FIONA Campaign at EUPHORE. In: Barnes, I., Rudziński, K. (eds), Disposal of Dangerous Chemicals in Urban Areas and Mega Cities. Role of Oxides and Acids of Nitrogen in Atmospheric Chemistry. NATO Science for Peace and Security Series C: Environmental Security. Springer, Dordrecht, 45-58, doi: 10.1007/978-94-007-5034-0_4, 2013.